# Applied machine learning for the risk-stratification and clinical decision support of hospitalised patients with dengue in Vietnam

**Damien K. Ming**[1]*, **Bernard Hernandez**[2,3], **Sorawat Sangkaew**[2], **Nguyen Lam Vuong**[4,5], **Phung Khanh Lam**[4,5], **Nguyen Minh Nguyet**[4], **Dong Thi Hoai Tam**[4], **Dinh The Trung**[4], **Nguyen Thi Hanh Tien**[4], **Nguyen Minh Tuan**[4,6], **Nguyen Van Vinh Chau**[4,7], **Cao Thi Tam**[7], **Ho Quang Chanh**[4,7], **Huynh Trung Trieu**[4,7], **Cameron P. Simmons**[8], **Bridget Wills**[4,9], **Pantelis Georgiou**[2,3], **Alison H. Holmes**[2], **Sophie Yacoub**[4,9], on behalf of the Vietnam ICU Translational Applications Laboratory (VITAL) investigators**

**1** Department of Infectious Disease, Imperial College London, United Kingdom, **2** Centre for Antimicrobial Optimisation, Imperial College London, United Kingdom, **3** Centre for BioInspired Technology, Imperial College London, United Kingdom, **4** Oxford University Clinical Research Unit, Centre for Tropical Medicine, Ho Chi Minh City, Vietnam, **5** University of Medicine and Pharmacy at Ho Chi Minh City, Ho Chi Minh City, Vietnam, **6** Children's Hospital 1, Ho Chi Minh City, Vietnam, **7** Hospital for Tropical Diseases, Ho Chi Minh City, Vietnam, **8** Institute of Vector Borne Disease, Monash University, Clayton, Australia, **9** Centre for Tropical Medicine and Global Health, Nuffield Department of Medicine, University of Oxford, United Kingdom

* d.ming@imperial.ac.uk

**Data Availability Statement:** The dataset used for analysis is available at the Oxford University

## Abstract

### Background

Identifying patients at risk of dengue shock syndrome (DSS) is vital for effective healthcare delivery. This can be challenging in endemic settings because of high caseloads and limited resources. Machine learning models trained using clinical data could support decision-making in this context.

### Methods

We developed supervised machine learning prediction models using pooled data from adult and paediatric patients hospitalised with dengue. Individuals from 5 prospective clinical studies in Ho Chi Minh City, Vietnam conducted between 12th April 2001 and 30th January 2018 were included. The outcome was onset of dengue shock syndrome during hospitalisation. Data underwent random stratified splitting at 80:20 ratio with the former used only for model development. Ten-fold cross-validation was used for hyperparameter optimisation and confidence intervals derived from percentile bootstrapping. Optimised models were evaluated against the hold-out set.

### Findings

The final dataset included 4,131 patients (477 adults and 3,654 children). DSS was experienced by 222 (5.4%) of individuals. Predictors were age, sex, weight, day of illness at hospitalisation, indices of haematocrit and platelets over first 48 hours of admission and before the onset of DSS. An artificial neural network model (ANN) model had best performance

Research Archive: https://doi.org/10.5287/bodleian:gAzqvApA4.

**Funding:** This work was supported by the Wellcome Trust grant [215010/Z/18/Z]. Authors DM and BH receive their salaries from and are supported by the grant. The funders had no role in study design, data collection and analysis, decision to publish, or preparation of the manuscript.

**Competing interests:** The authors have declared that no competing interests exist

with an area under receiver operator curve (AUROC) of 0.83 (95% confidence interval [CI], 0.76–0.85) in predicting DSS. When evaluated against the independent hold-out set this calibrated model exhibited an AUROC of 0.82, specificity of 0.84, sensitivity of 0.66, positive predictive value of 0.18 and negative predictive value of 0.98.

## Interpretation

The study demonstrates additional insights can be obtained from basic healthcare data, when applied through a machine learning framework. The high negative predictive value could support interventions such as early discharge or ambulatory patient management in this population. Work is underway to incorporate these findings into an electronic clinical decision support system to guide individual patient management.

## Introduction

Dengue is a systemic viral disease which exerts a significant health and economic burden worldwide. Up to 5% of hospitalised patients develop severe dengue, a life-threatening complication manifesting as shock, bleeding and/or organ dysfunction [1]. With an estimated 51 million symptomatic cases each year, seasonal epidemics and high caseloads impose a huge strain on local healthcare services [2]. The wide spectrum, and non-specific nature of clinical presentations pose further challenges to effective healthcare planning [3].

Strategies to identify patients who are at increased risk of complications such as dengue shock syndrome (DSS) during the early febrile phase of illness have been a subject of considerable research [4,5]. A widely-adopted approach particularly in low- and middle-income countries (LMICs) is the use of clinical warning signs outlined in the World Health Organisation (WHO) 2009 dengue guidelines [6]. The absence of these signs provides a high negative predictive value for severe dengue [7] and furthermore has relatively few requirements for implementation–needing only clinical examination findings and results from basic haematological tests. However, heterogeneity between care settings [8] and a suboptimal specificity have led to concerns that use of this system might not alleviate the excess, and potentially unnecessary hospitalisations when this system is adopted for admission triage [9]. In response, clinical prognostic models [10–12] and dengue-specific biomarkers [13,14] have been researched, but their limited performances or requirement for specialised laboratory testing represent barriers to widespread adoption. Tools for risk-stratification in dengue should: maximise the utility of existing patient data, be calibrated to local healthcare setting, and fit into the clinical decision-making workflow.

The rapid scale up of digital health developments, particularly within LMICs have enabled a data-driven approach previously not possible. In particular integration of electronic health record systems can strengthen healthcare systems [15] and provide the platforms to capitalise on novel data science methodologies such as machine learning. Machine learning is particularly suited for analysis of large scale, multidimensional data [16], as well as diverse data types such as physiological signals and radiological images utilised within healthcare. They provide empirical approaches to utilising data with reduced reliance on *a priori* assumptions. Within dengue research, machine learning methods have been applied in outbreak prediction and disease forecasting [17] but their role in clinical decision-making and risk stratification remains limited [18–20].

As part of a multi-disciplinary group for development of technological interventions in critical illness within LMIC settings, the Vietnam ICU Translational Applications Laboratory (VITAL), we adopted a data-driven approach to utilise supervised machine learning in a cohort of patients hospitalised with acute dengue to predict the development of DSS. We hypothesise that adopting such approaches on large, diverse datasets would offer a robust approach to apply findings from research data to real-world settings. The aim of this work is the development of a clinical decision-support system (CDSS) to serve as an adjunct in the clinical management of patients hospitalised with dengue.

## Methods

Results are reported following the TRIPOD Statement [21].

### Ethical approval

The study was approved by the scientific and ethical committee of the Hospital for Tropical Diseases (HTD), Ho Chi Minh City and by the Oxford Tropical Research Ethics Committee (OxTREC) with datasets pseudonymised prior to analyses (references 145–0420 and 146–0420).

### Source of data

The data used for this study comprises of an aggregation of prospective clinical studies conducted by Oxford University Clinical Research Unit (OUCRU) between 12th April 2001 and 30th January 2018 in healthcare facilities including the HTD within Ho Chi Minh City, Vietnam [10,12,22–24]. These included 4 prospective observational studies and 1 randomised control trial (ISRCTN39575233). Broadly, patients were eligible for inclusion if they presented with an acute febrile illness compatible with dengue on clinical assessment. Subsequent confirmation of dengue was done through one, or more of the following: i) a positive NS1 point of care assay or NS1 ELISA, ii) positive reverse transcriptase polymerase chain reaction (RT-PCR), iii) positive dengue IgM through acute serology, iv) or seroconversion of paired IgM samples. These criteria for dengue diagnosis are in line with standard study definitions employed in this setting. Further information on individual studies including recruitment criteria can be found in the S1 Appendix.

For patients enrolled, the decision for hospital admission was made according to the study inclusion criteria, and/or clinician assessment in line with national guidelines. Individuals managed in hospital include patients at higher risk of clinical deterioration but also those unsuitable for ambulatory management because of clinical and non-clinical factors. We defined adults as patients who were 18 years or above at time of enrolment.

Information common across studies including demographics, presenting features, investigations results and outcomes were extracted. In order to develop the prediction model, we excluded any observations obtained at the same time, or after the onset of DSS. For patients who did not develop DSS, observations obtained after 120 hours of illness onset were also excluded, where the onset of fever is taken as the start of illness. This timepoint represented the typical period of the critical phase and onset of shock in dengue. We included only patients who were randomised to the placebo arm for any interventional studies. In the final models patient predictive variables from all studies were measured within the first 48 hours of hospital admission–corresponding to a median of 4 and 5 days of illness and provides a useful timeframe for management in the early phase of dengue in hospital.

A diagram of patient inclusion is shown in Fig 1. Patient records were excluded from the final dataset given the following criteria–patients: i) with a final diagnosis of an illness other

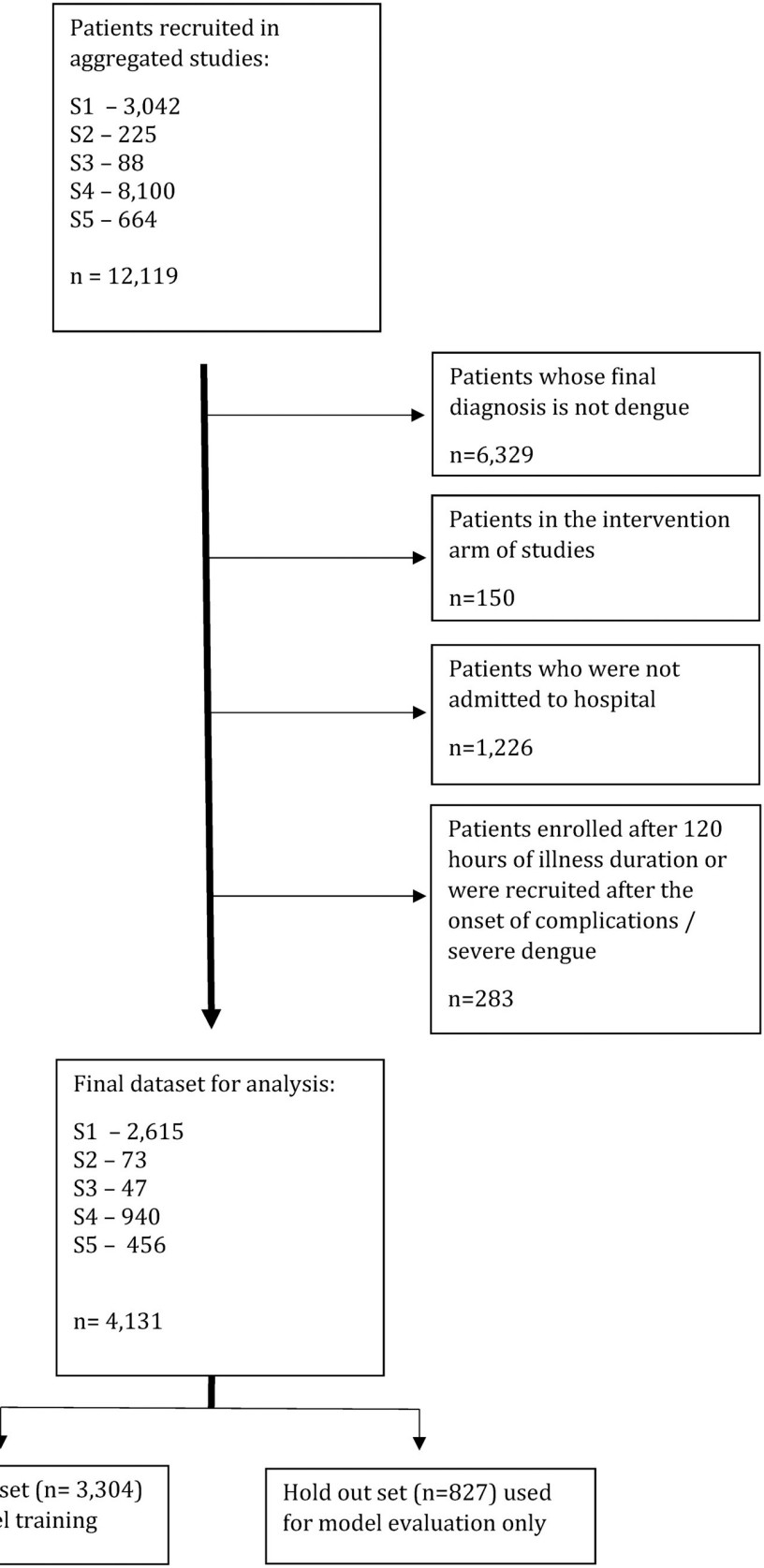

**Fig 1. CONSORT diagram for patients recruited in original clinical studies and processing to derive a development and hold out set for model evaluation**

than dengue (n = 6,329), ii) enrolled in the intervention arm of the randomised control trial (n = 150), iii) not hospitalised during illness (n = 1,226), iv) without data prior to day 5 of illness onset and where the development of DSS occurred at the same time, or before dates of predictor variables (n = 283). The final dataset consisted of 4,131 adult and paediatric patients.

## Prediction outcomes

The primary outcome was development of dengue shock syndrome (DSS) during hospital admission period as a binary classifier. DSS is defined as a pulse pressure equal to or less than 20 mmHg, or low blood pressure (BP) for age, with clinical signs of reduced peripheral perfusion [6]. A secondary analysis was also carried out with complicated dengue as the outcome of interest. This was defined as the presence of shock, significant bleeding and plasma leakage at any point during hospital stay (S1 Appendix).

## Predictors and missing data

Predictors used in the models include routine clinical and laboratory parameters, selected on the basis of expert consensus, data completeness and pragmatic utility suited to a LMIC healthcare setting. The final dataset for all models included predictors with less than 1.5% missing data. For each model, categorical data was fitted on the training set and imputed using the mode, and missing numerical data was imputed using both the mean and median and the method which resulted in better cross-validation performance was chosen for that particular model. Numerical features (age, day of illness, weight, haematocrit, platelet count) were either transformed by standardisation to achieve a mean value of 0 and standard deviation of 1, or left untransformed again depending on performance attained in cross-validation through a grid search process.

## Development and validation

The data underwent an initial random split using an 80/20 ratio to create a development and hold-out set using a stratified process to ensure proportional distribution of outcomes in both sets: the proportion of DSS in the development and hold-out sets were 5.4% and 5.3% respectively. The hold-out set was not used in any stage of the model development process and used only for testing of the final models.

## Model development process

Four diverse machine learning algorithms namely: extreme gradient boosting (XGBoost), random forest classifier, artificial neural network and support vector machines were used for model development. Logistic regression using lasso or ridge regression was also used as a baseline comparator.

Each algorithm was trained using the development set only through stratified k-fold cross-validation in order to establish optimal hyperparameters whilst minimising overfitting. The development set was split into 10 equal-sized folds and models with specific hyperparameter configurations trained on the 9 folds, and validated against the remaining fold. This process was then repeated 10 times with a different fold in each iteration. A grid-search process was used to generate different combinations of hyperparameters for each classification algorithm. The area under the receiver operating curve (AUROC) was used for scoring, and other

reported metrics included sensitivity, specificity, positive predictive value (PPV), negative predictive value (NPV) and calibration using the Brier score. Confidence intervals for model performance metrics were estimated through bootstrap resampling of the development set and tested against the out of bag samples, repeated 1,000 times. We addressed the class imbalance in the dataset through Synthetic Minority Over-sampling Technique (SMOTE) within our pipelines. A probabilistic classifier method was used to compare predicted and actual outcomes. The optimal cut-off value was determined according to Youden's J-statistic for each algorithm in order to provide best all-around performance. Models were calibrated through isotonic calibration through a 10-fold cross validation process.

Final models for each of the algorithms were then tested with the hold-out set, in order to describe performance against an independent, previously unseen dataset. A Shapley Additive exPlanations (SHAP) analysis was performed to analyse models for interpretability. This utilises an independent method to evaluate the impact of individual predictors within the model with regards to predicted outcomes: a greater SHAP value denotes a higher contribution towards a prediction of complicated dengue, taking into account interaction effects. References and the model development process are detailed in S1 Appendix. All analyses were performed in Python 3.7.

## Results

### Description

The final dataset included 4,131 patients hospitalised between 12th April 2001 and 30th January 2018 from 5 studies. Baseline patient characteristics are shown in Table 1. The number of female patients was 2,008 (49%) and the median age was 12 years old (interquartile range, IQR 9–14 years). The median day of illness on presentation to hospital was 3 days (IQR 2–4 days). In total, 222 (5.4%) patients experienced DSS during hospitalisation of which 209 (94.1%) were children and 13 (5.9%) were adults.

### Model predictors

Final predictors included in the model are patient age, sex, weight, day of illness onset on hospital admission, and the minimum, median and maximum haematocrit and platelet count

**Table 1. Baseline characteristics of individual studies included in the final dataset.**

| Study | Recruitment dates | Patients enrolled (n) | Patients with confirmed diagnosis of dengue (n) | Patients included in analyses (n) | Patients included in analyses | | | | | |
|---|---|---|---|---|---|---|---|---|---|---|
| | | | | | Age (median, IQR) | Day of illness onset on hospitalisation (median, IQR) | Female sex (n) | Secondary infection (n) | Shock (n) | Outcome measurement |
| S1 | 12/4/2001–24/7/2009 | 3,042 | 3,042 | 2,615 | 12 (10–13) | 4 (3–4) | 1,079 (41%) | 1,536/1,903 (81%) | 170 (7%) | WHO 1997 dengue outcomes |
| S2 | 3/8/2009–8/12/2010 | 225 | 225 | 73 | 15 (13–20) | 1 (1–2) | 18 (25%) | 42/64 (66%) | 3 (4%) | WHO 2009 dengue outcomes |
| S3 | 8/12/2010–16/6/2011 | 88 | 88 | 47 | 23 (21–25) | 3 (3–3) | 28 (40%) | 11/38 (29%) | 8 (17%) | WHO 2009 dengue outcomes |
| S4 | 19/10/2010–4/12/2014 | 8,100 | 2,245 | 940 | 12 (10–16) | 2 (1–3) | 427 (45%) | 687/915 (75%) | 31 (3%) | WHO 2009 dengue outcomes |
| S5 | 20/10/2016–30/1/2018 | 664 | 542 | 456 | 30 (27–45) | 4 (3–5) | 456 (100%) | 494/539 (92%) | 10 (2%) | WHO 2009 dengue outcomes |

**Table 2. Characteristics of cohort (n = 4,131) divided by complication outcome during hospital. Haematological values and ranges refer to results taken over the initial 48 hours of hospitalisation only.** Data is presented as median and brackets denote the interquartile range. Univariate analyses were done using the Mann-Whitney test or Chi Squared test as appropriate.

| | Shock n = 222 | No shock n = 3,909 | p-value | Missing data (%) |
|---|---|---|---|---|
| | Median (IQR) | Median (IQR) | | |
| Median age (years) | 11 (9–13) | 12 (9–14) | 0.07 | 1.5 |
| Median day of illness at hospital admission (days) | 3 (2–4) | 3 (2–4) | 0.05 | 0 |
| Median weight (kg) | 34 (27–43) | 36 (27–45) | 0.26 | 0.9 |
| Female sex (%) | 90/222 (41%) | 1,918/3,909 (49%) | 0.02 | 0 |
| Median haematocrit (%) | 41 (39–43) | 40 (37–42) | <0.001 | 0.2 |
| Maximum haematocrit (%) | 48 (44–51) | 42 (39–45) | <0.001 | 0.2 |
| Minimum haematocrit (%) | 37 (35–39) | 38 (35–40) | 0.05 | 0.2 |
| Maximum platelet count (x $10^9$/L) | 140 (102–192) | 159 (118–207) | <0.001 | 0.2 |
| Median platelet count (x $10^9$/L) | 70 (45–99) | 116 (83–159) | <0.001 | 0.2 |
| Minimum platelet count (x $10^9$/L) | 29 (18–58) | 86 (51–135) | <0.001 | 0.2 |

obtained over the first 48 hours of hospital admission. There were significant differences in distributions of sex, day of illness on hospital admission and indices of haematocrit and platelets values between patients with, or without DSS. Characteristics of the two groups and the comparisons of predictors are shown in Table 2.

## Model evaluation

Univariate analysis between the development (n = 3,304) and hold-out (n = 837) set showed no significant statistical differences across included predictors. In cross-validation of the development dataset, all machine learning algorithms used in this study resulted in similar mean AUROC (between 0.77 and 0.83). The performance of logistic regression using linear terms resulted in an AUROC of 0.79 (95% CI 0.74–0.83).

The artificial neural network (ANN) classifier model provided the best discrimination performance in predicting DSS, with an AUROC of 0.83 (95% CI, 0.76–0.85). When optimal cut-off thresholds were chosen using the J-statistic, this model provided a specificity of 0.88 (0.68–0.93), sensitivity of 0.66 (0.52–0.81) PPV of 0.24 (0.12–0.33), NPV of 0.98 (0.97–0.99) and a Brier score of 0.07. The performance results of all optimised final models from cross-validation using development data are displayed in Table 3.

Sensitivity analyses exploring performance of the model to classify complicated dengue as an endpoint, contribution of batch effects, WHO classifications and performance of the model stratified by age (paediatric or adult cohort) are shown in the S1 Appendix.

**Table 3. Performance of final models for each algorithm with respect to internal 10-fold cross validation on the development set (n = 3,304).** The 95% confidence intervals shown in brackets were derived from bootstrap resampling of the development and testing against out of bag samples, repeated 1,000 times.

| Model | Mean AUROC | Specificity | Sensitivity | Positive Predictive value | Negative Predictive value | Brier score |
|---|---|---|---|---|---|---|
| XGBoost | 0.81 (0.74–0.84) | 0.90 (0.70–0.93) | 0.59 (0.52–0.78) | 0.26 (0.12–0.31) | 0.97 (0.97–0.98) | 0.068 |
| Random forest | 0.80 (0.75–0.84) | 0.82 (0.75–0.93) | 0.71 (0.54–0.76) | 0.18 (0.14–0.31) | 0.98 (0.97–0.98) | 0.068 |
| Logistic regression | 0.79 (0.74–0.83) | 0.89 (0.71–0.91) | 0.62 (0.56–0.78) | 0.24 (0.13–0.28) | 0.98 (0.97–0.98) | 0.075 |
| Artificial neural network | 0.83 (0.76–0.85) | 0.88 (0.68–0.93) | 0.66 (0.52–0.81) | 0.24 (0.12–0.33) | 0.98 (0.97–0.99) | 0.068 |
| Support vector machines* | 0.82 (0.75–0.84) | 0.86 (0.68–0.92) | 0.66 (0.53–0.82) | 0.21(0.11–0.29) | 0.98 (0.97–0.99 | 0.13 |

*implementations of SVM were done with SMOTE.

**Table 4. Performance metrics of final calibrated models when evaluated against the hold-out set (n = 827).**

| Model | Mean AUROC | Specificity | Sensitivity | Positive Predictive value | Negative Predictive value | Brier score |
|---|---|---|---|---|---|---|
| XGBoost | 0.85 | 0.91 | 0.64 | 0.29 | 0.98 | 0.04 |
| Random forest | 0.84 | 0.87 | 0.68 | 0.23 | 0.98 | 0.04 |
| Logistic regression | 0.79 | 0.90 | 0.55 | 0.24 | 0.97 | 0.04 |
| Artificial neural network (ANN) | 0.82 | 0.84 | 0.66 | 0.18 | 0.98 | 0.04 |
| Support vector machines | 0.83 | 0.84 | 0.66 | 0.19 | 0.98 | 0.04 |

## Model evaluation against hold-out test set

Final models were evaluated against the independent hold-out set of 827 patients not involved in development. The XGBoost model provided the highest discrimination performance (AUROC 0.85, specificity 0.91, sensitivity 0.64, PPV of 0.29 and NPV of 0.98), although discrimination across other machine learning classifiers were similar (AUROC ranging from 0.79 to 0.85). Logistic regression model performance in this setting provided an AUROC of 0.79 (Table 4).

## Model interpretability

To provide interpretability of the XGBoost model we performed a SHAP analysis. A summary plot (Fig 2) shows that within the model a higher maximum haematocrit, a lower minimum platelet count and female sex and lower age contributed towards complicated dengue prediction with a non-linear relationship between SHAP values and individual predictor variables (Fig 3). SHAP values for individual features for the other models are shown in the S1 Appendix. All models ranked maximum haematocrit and minimum platelet count highest in terms of the importance of features as characterised by SHAP.

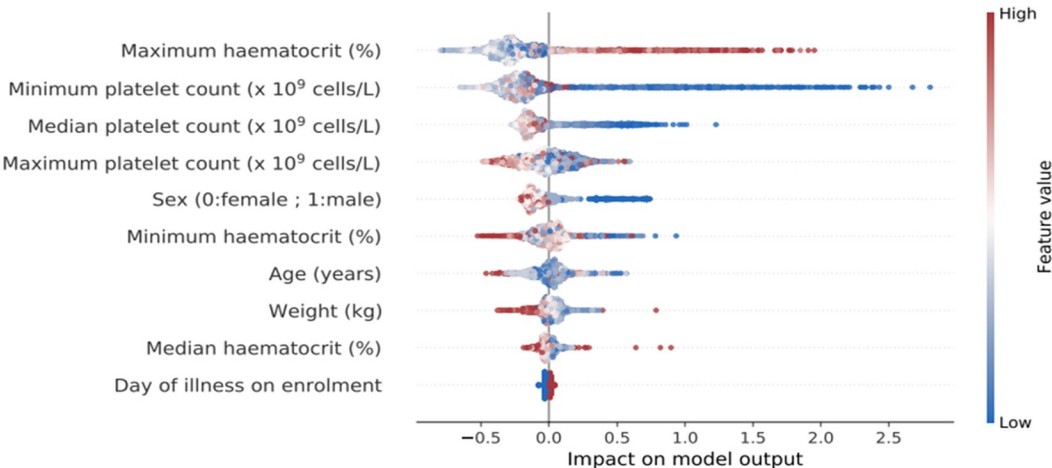

**Fig 2. Summary plot of SHAP values for the XGBoost model.** The plot shows the contribution of individual predictors and their range of values towards final model output prediction, where shock and no shock are represented by 1 and 0 on the x-axis respectively. The main predictors are arranged in descending importance for the model. The colours of the individual features represent whether the values are high or low. For example, a higher maximum haematocrit (red) is associated with a positive impact on model output and thus associated with dengue shock. Female sex is represented by blue (0) and male sex represented by red (1).

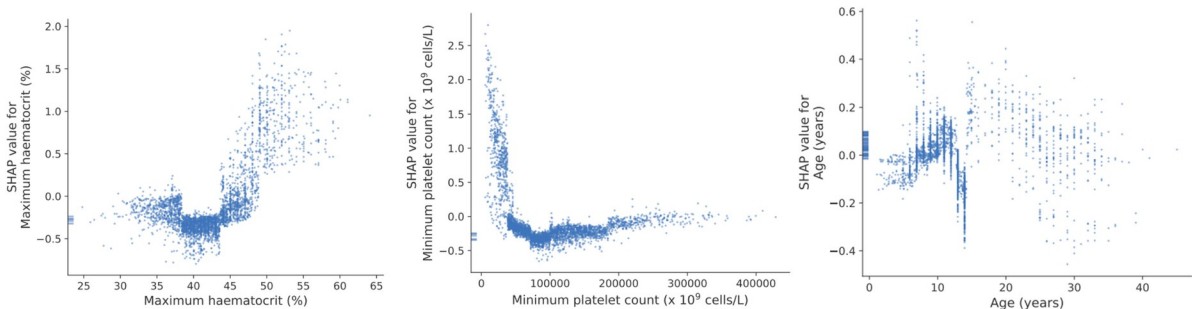

**Fig 3.** Scatter plot of predictor values (x-axis) against SHAP values (y-axis) in the XGBoost model for haematocrit, platelet count and age. The plot shows a non-linear relationship between predictor values and model output.

## Discussion

We trained and evaluated machine learning models to predict onset of dengue shock syndrome in hospitalised patients in Vietnam, using only basic demographic and laboratory predictors measured during the first 48 hours of hospital admission. These models were able to robustly model the non-linear relationships between predictors and outcome, with the ANN algorithm demonstrating optimal discrimination in terms of AUROC in cross-validation and when evaluated against the independent hold-out set.

Our models take into account changes in haematocrit and platelet over the first 48 hours of admission as predictors and classifies subsequent risk of DSS. The negative predictive value of 0.98 provided by the ANN model may be of clinical utility by allowing an automatic identification of admitted patients who are at lower risk of developing complications. The performance metrics of our machine learning model is commensurate with those of other prognostic models [18] although direct comparisons can be difficult given differences in practice and care pathways.

Patients with dengue in our setting are admitted to hospital for a variety of indications: when clinicians consider them at higher risk, have certain comorbidities, or in presence of non-medical factors such as logistical challenges for outpatient follow up. We plan to implement these findings through a clinical decision support system (CDSS) interface developed by our group, and previously validated in real-world clinical settings [25]. This web-based system allows the model to be run on most computers or tablets without any specialised requirements. Usability and uptake are important for implementation within clinical settings. Studies characterising workflow, end-user requirements and optimal interface designs are currently taking place at our hospital to optimise local roll-out and utility in the first instance. The feasibility of applying these models to other regions with endemic dengue has been shown to be possible [26], but factors such as differences in local healthcare processes, treatment guidelines and access to healthcare have to be taken into account.

The inclusion of only basic predictors for our model, namely age, sex, weight, haematocrit and platelet concentrations were prioritised. These measurements are readily-accessible in many LMIC care settings and are known significant factors in dengue risk stratification [12,27]. Although overall performance of our optimal model may not supersede that of clinical assessment, its implementation within a CDSS environment in hospitals could provide an additional layer of patient safety and consistency in care. It could also support interventions such as early hospital discharge and/or evaluate the suitability for ambulatory management within this group. The role of such a CDSS is not to take away from clinicians' assessment but one which supports and complements their workflow [28]. Ensuring that the model is updated

with available new data will be essential such as use of an iterative system of training and evaluation over time. A prospective validation study is currently underway at our institution to examine the real-world performance of these models. In their development we used the J-statistic to optimise all-round performance but specific metrics of the model, such as its negative predictive value, will need to maintained by threshold adjustment in order for this to be clinically relevant. This threshold is likely dynamic and will depend on factors including seasonality and hospital caseload.

This study provides a proof-of-principle and baseline that basic healthcare data can offer added-value when processed through novel data-science methods such as machine learning. These methods are particularly suited for data without a clear underlying linear relationship or that which are prone to noise or artifact, including continuous physiological data. For example, photoplethysmography signals used in pulse oximetry is predictive for dengue shock analysed through machine learning [29]. Capture of such data through low-cost (<$100 USD) devices or wearable technologies [30], and their incorporation into predictive algorithms is an area of active area interest.

The strengths of our study include the use of a prospectively-collected set of clinical data for machine learning model development and validation. This is the largest sample size used for this purpose to our knowledge and increases the prospective generalisability of our findings. Minimal missing data (<1.5%) reduced requirements for imputation also increases validity of the results. For model development, the use of cross-validation and an independent hold-out set are robust methods of reducing overfitting, impact of noise and issues regarding heterogeneity within the data.

There were limitations to our study. We used data aggregated from five different clinical studies conducted at OUCRU over 19 years to build a large training cohort. Inherent differences in study design between datasets, such as the threshold for hospital admission or pregnancy status exist, leading to underlying heterogeneity. Although the overall inclusion criteria for patients into the analysis were comparable i.e., hospitalised patients with dengue during the febrile phase–underlying biases might exist which affect future performance. There are intrinsic differences between each clinical study as they address specific research questions and span many years not amenable to standard normalisation methods (S1 Appendix). We have therefore not explicitly corrected for these differences prior to dataset aggregation, in part to also retain variance needed for prospective generalisability and real-world performance. Whether this approach is optimal will be ultimately dependent on performance of the model on an independent, prospective cohort of patients.

We adopted DSS as the primary outcome as this was captured most accurately within the datasets. This is a relatively rare outcome, leading to an imbalanced dataset and affects model calibration. It is also acknowledged that other clinical features of moderate and severe dengue (such as bleeding and organ impairment) have not been considered in the primary model—we therefore developed a secondary model examining the association of complicated dengue with feature variables (S1 Appendix) and show similar performances suitable for implementation. Dengue shock and severe disease can also present earlier in illness prior to hospitalisation and therefore this model might have limited utility for this subset of patients.

Only a limited number of features collected as part of routine clinical care were included in our final models and as a result feature selection/ engineering was not possible. It is possible that relevant clinical features such as dengue serotype or indices of viral load could result in better model performance although this has to be balanced with feasibility of model implementation across LMIC clinical settings. Finally, this was a predominantly paediatric patient training cohort, and model performance was less robust when tested against older patients. Future iterations of the model would need to take into account optimising performance to ensure fairness across different age groups.

In conclusion, we present results from a machine learning approach to predict the risk of dengue shock in hospitalised patients with dengue. We demonstrate performance metrics suitable for clinical evaluation and plan prospective studies to understand if these methods, when delivered through a decision support tool, can translate into improvements in clinical care.

## Supporting information

**S1 Appendix. Description of data source and individual studies, description of data source and individual studies and secondary and sensitivity analyses.**
(DOCX)

## Acknowledgments

The authors would like to acknowledge the patients who participated, the doctors and nurses who cared for the patients, and the laboratory staff at the Hospital for Tropical Diseases.

## Author Contributions

**Conceptualization:** Damien K. Ming, Bernard Hernandez, Sophie Yacoub.

**Data curation:** Damien K. Ming, Bernard Hernandez.

**Formal analysis:** Damien K. Ming, Bernard Hernandez, Sorawat Sangkaew.

**Investigation:** Damien K. Ming, Nguyen Lam Vuong, Nguyen Minh Nguyet, Dong Thi Hoai Tam, Dinh The Trung, Nguyen Thi Hanh Tien, Nguyen Minh Tuan, Nguyen Van Vinh Chau, Cao Thi Tam, Huynh Trung Trieu, Cameron P. Simmons, Bridget Wills.

**Methodology:** Damien K. Ming, Bernard Hernandez, Sorawat Sangkaew, Phung Khanh Lam, Cameron P. Simmons, Bridget Wills.

**Project administration:** Damien K. Ming.

**Software:** Bernard Hernandez.

**Supervision:** Bernard Hernandez, Cameron P. Simmons, Bridget Wills, Pantelis Georgiou, Alison H. Holmes, Sophie Yacoub.

**Writing – original draft:** Damien K. Ming, Bernard Hernandez, Sorawat Sangkaew, Sophie Yacoub.

**Writing – review & editing:** Damien K. Ming, Bernard Hernandez, Sorawat Sangkaew, Nguyen Lam Vuong, Phung Khanh Lam, Nguyen Minh Nguyet, Dong Thi Hoai Tam, Dinh The Trung, Nguyen Thi Hanh Tien, Nguyen Minh Tuan, Nguyen Van Vinh Chau, Cao Thi Tam, Ho Quang Chanh, Huynh Trung Trieu, Cameron P. Simmons, Bridget Wills, Pantelis Georgiou, Alison H. Holmes, Sophie Yacoub.

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
