## [Decision Letter · Decision Letter 0]

7 Sep 2021

PDIG-D-21-00002

Applied machine learning for the risk-stratification and clinical decision support of hospitalised patients with dengue in Vietnam

PLOS Digital Health

Dear Dr. Ming,

Thank you for submitting your manuscript to PLOS Digital Health. After careful consideration, we feel that it has merit but does not fully meet PLOS Digital Health’s publication criteria as it currently stands. Therefore, we invite you to submit a revised version of the manuscript that addresses the points raised during the review process.

EDITOR: Please insert comments here and delete this placeholder text when finished. Be sure to:

Indicate which changes you require for acceptance versus which changes you recommendAddress any conflicts between the reviews so that it's clear which advice the authors should followProvide specific feedback from your evaluation of the manuscript

Please ensure that your decision is justified on PLOS Digital Health’s publication criteria and not, for example, on novelty or perceived impact.

We look forward to receiving your revised manuscript.

Kind regards,

Ryan S McGinnis, Ph.D.

Academic Editor

PLOS Digital Health

Journal Requirements:

1. Please provide separate figure files in .tif or .eps format only, and remove any figures embedded in your manuscript file.  If you are using LaTeX, you do not need to remove embedded figures.

For more information about figure files please see our guidelines: https://journals.plos.org/digitalhealth/s/figures

2. Please amend your detailed Financial Disclosure statement. This is published with the article, therefore should be completed in full sentences and contain the exact wording you wish to be published.

i). Please include all sources of funding (financial or material support) for your study. List the grants (with grant number) or organizations (with url) that supported your study, including funding received from your institution. 

ii). State the initials, alongside each funding source, of each author to receive each grant.

iii). State what role the funders took in the study. If the funders had no role in your study, please state: “The funders had no role in study design, data collection and analysis, decision to publish, or preparation of the manuscript.”

iv). If any authors received a salary from any of your funders, please state which authors and which funders.

3. Please update the completed 'Competing Interests' statement, including any COIs declared by your co-authors. If you have no competing interests to declare, please state "The authors have declared that no competing interests exist". Otherwise please declare all competing interests beginning with the statement "I have read the journal's policy and the authors of this manuscript have the following competing interests:

4. In the online submission form, you indicated that "The original clinical datasets used contains identifying or sensitive patient information relating to past and ongoing clinical studies and access is subject to research ethics committee approval from HTD ethical committee and OxTREC approval. Please contact the authors for further information and data requests.". All PLOS journals now require all data underlying the findings described in their manuscript to be freely available to other researchers, either 1. In a public repository, 2. Within the manuscript itself, or 3. Uploaded as supplementary information.

Additional Editor Comments (if provided):

The three reviewers have seen merit in this work, but also identify several areas that could be improved to strengthen the manuscript. In particular, and in keeping with the aims of the journal, the authors are encouraged to make their code and data available.

Reviewers' comments:

Reviewer's Responses to Questions

**Comments to the Author**

1. Does this manuscript meet PLOS Digital Health’s publication criteria? Is the manuscript technically sound, and do the data support the conclusions? The manuscript must describe methodologically and ethically rigorous research with conclusions that are appropriately drawn based on the data presented.

Reviewer #1: Partly

Reviewer #2: Yes

Reviewer #3: Yes

2. Has the statistical analysis been performed appropriately and rigorously?

Reviewer #1: Yes

Reviewer #2: Yes

Reviewer #3: Yes

3. Have the authors made all data underlying the findings in their manuscript fully available (please refer to the Data Availability Statement at the start of the manuscript PDF file)?

Reviewer #1: No

Reviewer #2: No

Reviewer #3: No

4. Is the manuscript presented in an intelligible fashion and written in standard English?

Reviewer #1: Yes

Reviewer #2: Yes

Reviewer #3: Yes

5. Review Comments to the Author

Reviewer #1: The paper presents the use of machine learning algorithms to predict dengue shock syndrome based on an integrated dataset originating from 5 different studies. Overall, the research methodology and the methods are well presented. The issues with the work are as follows: (i) The selection of the predictors is opportunistic and hence it is unclear whether there are other clinical features that should have been included. The list of predictors is quite simplistic from a clinical perspective, though the positive is that these predictors still are given high prediction accuracy; (ii) Given the limited number of predictors, there is no opportunity to perform feature selection which is a limitation; (iii) Correlation between the features and the outcome should be checked, and confounders should be established; (iv) The prediction models perform binary classification, where there is a significant class imbalance such that only 5.4% of patients had a positive dengue shock. The study does not use any methods to address the class imbalance, and it is interesting to note such a high sensitivity reported in the results despite the class imbalance. Furthermore, which such a low number of the dengue class the authors should point out the ratio of the two classes in both the training and evaluation sets. the study can be improved by performing data augmentation using GAN or simply using SMOTE and then compare results between the original and the augmented dataset; (v) Table 1: is there an explanation for why the dengue shock numbers are so high in cohort S1 despite cohort S4 has a higher number of patients. It seems that there may be some issues with the inclusion criteria that need proper investigation; (vi) The aggregation of data from multiple temporal and institutional cohorts should take into account the clinical interpretation of dengue shock across these institutions to have a standardized inclusion criteria. It would be useful to develop independent models for each cohort to investigate the influence of the different predictors across cohorts; (vii) The data spans 19 years which is a rather long period given improvements in diagnostic methods and treatments--as the authors suggest that this study points to the utility of the model in LMIC given the need for a limited number of clinical attributes, it will be useful to stratify the data into temporal periods and then develop prediction models for each period to confirm that the predictors hold there importance across time; (viii) The data does not distinguish between paediatric patients and adults/elderly patients despite both groups have different clinical characteristics, especially elderly having co-morbidities. Therefore, having a universal model for all age groups is susceptible to bias towards a specific age group. (ix) The authors should provide details of how the trained models were calibrated; (x) The discussion should include insights into the clinical pragmatics of this work, the actual design and implementation of the digital health based solution in a clinical workflow, alluding to data access interfaces, etc. Also, the clinical setting in which it will be used and how will its use be determined on a per patient basis; (xi) The authors should comment on the scalability of this model to other regions; (xii) What plans are in place to keep the model current given the availability of new data and changes in clinical processes;

The paper does not present any novel work with regards to methods--it reports the application of existing machine learning methods to a curated dataset. So the original contribution of the paper basically is the preparation of the dataset which is insufficient. The authors should highlight any novelty of their work in terms of methods and outcomes.

Reviewer #2: The authors have ~4000 patients from 5 different data sources, and are trying to make different models (XGBoost, Logistic Regression, ANN, Random forest and SVM) to classify the patients according to if they have DSS (Dengue shock syndrome) they also do a secondary analysis making models for complications in DSS they have also done SHAP interpretability analysis for xgboost and logistic regression. Overall, the paper was well written we have some points that we found could be beneficial to improve the quality of the work

Major comments:

Authors should mention the effect of different data sources, i.e. data batches. Specifically, if the sampling strategy employed addresses potential biases due to discontinous data collection in batches over the years. technologies/approaches must have changed over the long period of collection. It is important to account for these batch effects as they may dramatically change the outcome.

As a suggestion, to reduce the bias in the model caused by data coming from 5 sources, and subsequently increase its generalization performance, the authors could have normalized the data from each source separately and then concatenated these 5 normalized datasets into one, instead of first concatenating and then normalizing. The authors haven’t mentioned when they performed the normalizing step but since they

mentioned it at the end in line 177, it is reasonable to assume that it was the last step. Having clarity on how this normalization was performed and at which step during the process is important.

Also, the entire analysis and the deidentified data should be hosted on a public platform such as github to allow researchers to reproduce the work.

Minor comments:

1) In Abstract Findings : The authors should give the distribution of patients who experienced DSS (how many of them were adults and how many were children)

2) Line 146: The authors should mention what symptoms they have taken to be ‘onset of illness’ in patients who did not develop DSS.

3) Line 177: Predictors used in the model which were transformed should be mentioned.

4) The authors could specify the Brier scores for evaluation against the hold out set for the primary analysis too, as they have done for the secondary analysis (Supplementary Appendix 3)

5) Supplementary Appendix 2: In the hyperparameters of the XGBoost model, the authors mention 2 optimal models. The authors should specify if these have equal AUROC, or if one was chosen because of less overfitting and other for better performance, or whatever else the case might be. Also, the ‘min_child_weight’ in both of these is different from the set of values they have taken initially (0.005 and 0.001 are not present in [0.05, 0.1, 0.2])

6) Supplementary Appendix 2: As a side note, the authors have taken very few parameters into account for the XGBoost model, and it could be possible to achieve a better AUROC using hyperparameters like ‘subsample’, ‘colsample_by...’ and ‘alpha’ and using a more extensive set of values for parameters like ‘max_depth’ and especially ‘min_child_weight’, for which the values used are quite small in magnitude.

7) Supplementary Appendix 2 (Interpretability and SHAP analysis): SHAP is only applied for XGBoost and logistic regression. The authors can use SHAP’s KernelExplainer to explain any model. However, making the MLP(ANN) using PyTorch/Tensorflow/keras and then using SHAP’s DeepExplainer or GradientExplainer classes will lead to better explanations and interpretability.

Reviewer #3: It was a delight to review this paper, which in its present form I could readily imagine appearing in the journal. I only have a few comments for the authors to address.

L174: Provide more information on how imputation was done. I can’t work out from this description how the mean or median was selected.

L176: Please provide a better explanation of which data were standardised and how the decision was made to standardise. Don’t some of the algorithms (such as lasso) automatically standardise covariates?

L202: I was surprised to see the optimal cut off being determined through a statistical algorithm rather than through the clinical requirements. I was expecting the decision to be based on, for instance, a maximum tolerable number of patients with DSS being sent home (presumably to die) or the hospital capacity, rather than Youden’s J statistic.

L237: Was stochastic domination assured for these p-values?

L260: What is linear logistic regression?

L269 (figure 2): This is very pretty but probably requires a bit more information to guide readers’ interpretation. For instance, with sex, how do we interpret the different shades of blue and red and white? Also I wonder whether the plot should be taller so that the densities are easier to perceive.

L316: Please check and confirm that the Tan Tock Seng Hospital (Singapore) dengue clinical cohort is not larger. I had the impression it was similar in size, and they’ve used it for similar applications, though obviously their patients are quite different.

Data availability: It would be preferable if a sanitised version of the dataset could be provided, with just the variables that were included in the final analysis so no need for information that could identify the patients.

6. PLOS authors have the option to publish the peer review history of their article (what does this mean?). If published, this will include your full peer review and any attached files.

**Do you want your identity to be public for this peer review?** For information about this choice, including consent withdrawal, please see our Privacy Policy.

Reviewer #1: No

Reviewer #2: No

Reviewer #3: No

---

## [Decision Letter · Decision Letter 1]

15 Nov 2021

Applied machine learning for the risk-stratification and clinical decision support of hospitalised patients with dengue in Vietnam

PDIG-D-21-00002R1

Dear Dr. Ming,

We're pleased to inform you that your manuscript has been judged scientifically suitable for publication and will be formally accepted for publication once it meets all outstanding technical requirements.

Within one week, you'll receive an e-mail detailing the required amendments. When these have been addressed, you'll receive a formal acceptance letter and your manuscript will be scheduled for publication.

An invoice for payment will follow shortly after the formal acceptance. To ensure an efficient process, please log into Editorial Manager at https://www.editorialmanager.com/pdig/ click the 'Update My Information' link at the top of the page, and double check that your user information is up-to-date. If you have any billing related questions, please contact our Author Billing department directly at authorbilling@plos.org.

Kind regards,

Ryan S McGinnis, Ph.D.

Academic Editor

PLOS Digital Health

Additional Editor Comments (optional):

Reviewers' comments:

Reviewer's Responses to Questions

**Comments to the Author**

1. If the authors have adequately addressed your comments raised in a previous round of review and you feel that this manuscript is now acceptable for publication, you may indicate that here to bypass the “Comments to the Author” section, enter your conflict of interest statement in the “Confidential to Editor” section, and submit your "Accept" recommendation.

Reviewer #3: All comments have been addressed

2. Does this manuscript meet PLOS Digital Health’s publication criteria? Is the manuscript technically sound, and do the data support the conclusions? The manuscript must describe methodologically and ethically rigorous research with conclusions that are appropriately drawn based on the data presented.

Reviewer #3: Yes

3. Has the statistical analysis been performed appropriately and rigorously?

Reviewer #3: Yes

4. Have the authors made all data underlying the findings in their manuscript fully available (please refer to the Data Availability Statement at the start of the manuscript PDF file)?

Reviewer #3: Yes

5. Is the manuscript presented in an intelligible fashion and written in standard English?

Reviewer #3: Yes

6. Review Comments to the Author

Reviewer #3: Thanks for addressing my previous points. It would be good at the copy editing stage to give the doi of the data uploaded to ORA to make it easier to find.

7. PLOS authors have the option to publish the peer review history of their article (what does this mean?). If published, this will include your full peer review and any attached files.

**Do you want your identity to be public for this peer review?** For information about this choice, including consent withdrawal, please see our Privacy Policy.

Reviewer #3: No
